# Secreted Secondary Metabolites Reduce Bacterial Wilt Severity of Tomato in Bacterial–Fungal Co-Infections

**DOI:** 10.3390/microorganisms9102123

**Published:** 2021-10-09

**Authors:** Nandhitha Venkatesh, Max J. Koss, Claudio Greco, Grant Nickles, Philipp Wiemann, Nancy P. Keller

**Affiliations:** 1Department of Plant Pathology, University of Wisconsin—Madison, Madison, WI 53706, USA; thiruvannama@wisc.edu; 2Department of Medical Microbiology and Immunology, University of Wisconsin—Madison, Madison, WI 53706, USA; mjkoss@wisc.edu (M.J.K.); claudio.greco@jic.ac.uk (C.G.); gnickles@wisc.edu (G.N.); philipp.wiemann@solugentech.com (P.W.); 3Department of Bacteriology, University of Wisconsin—Madison, Madison, WI 53706, USA

**Keywords:** secondary metabolites, plant–microbe interactions, coinfection, wilt disease, bacterial–fungal interactions, *Fusarium oxysporum*, *Ralstonia solanacearum*

## Abstract

In order to gain a comprehensive understanding of plant disease in natural and agricultural ecosystems, it is essential to examine plant disease in multi-pathogen–host systems. *Ralstonia*
*solanacearum* and *Fusarium oxysporum f. *sp.* lycopersici* are vascular wilt pathogens that can result in heavy yield losses in susceptible hosts such as tomato. Although both pathogens occupy the xylem, the costs of mixed infections on wilt disease are unknown. Here, we characterize the consequences of co-infection with *R. solanacearum* and *F. oxysporum* using tomato as the model host. Our results demonstrate that bacterial wilt severity is reduced in co-infections, that bikaverin synthesis by *Fusarium* contributes to bacterial wilt reduction, and that the arrival time of each microbe at the infection court is important in driving the severity of wilt disease. Further, analysis of the co-infection root secretome identified previously uncharacterized secreted metabolites that reduce *R. solanacearum* growth in vitro and provide protection to tomato seedlings against bacterial wilt disease. Taken together, these results highlight the need to understand the consequences of mixed infections in plant disease.

## 1. Introduction

The soil microbiome plays a vital role in plant health [1] and microbial communication greatly alters disease outcomes [2]. The co-occurrence and interactions between soil microbes and their hosts fall on a spectrum—from mutualism, competition, or antagonism to simple coexistence—all mediated by direct interactions with each other and/or host-mediated responses [3]. Metagenomic studies have enabled the identification of several co-infection pathosystems where the multi-pathogen–host interactions alter pathogen growth and fitness [4,5] as well as epidemiological outcomes and virulence evolution [6]. Further, mechanistic insights on pathogen communication and changes in host response during co-infection can inform creative and novel disease management strategies. These examples emphasize the need to study the role of microbial interactions in plant diseases within multi-pathogen-host systems.

The challenges of mixed infections in human disease have been increasingly documented in recent years [7,8] and are further highlighted by recent lethal combinations of COVID-19 and other infections [9,10]. However, this topic still remains in its infancy in plant disease. *Ralstonia solanacearum* and *Fusarium oxysporum,* bacterial and fungal plant pathogens that cause wilt disease, can lead to yield losses up to 90% [11] and 80%, respectively, in a wide range of hosts [12,13]. While these pathogens have been studied in single pathogen–host systems, consequences of mixed infections are unknown. Previous studies from our lab showed that *R. solanacearum* can invade fungal chlamydospores and induce the production of antibacterial mycotoxins by *Fusarium* spp. [14]. The two pathogens significantly overlap in their presence across the world (based on CABI distribution data [15,16]) as their optimal growth temperatures are between 28–30 °C and the microbes display similar infection routes and colonize xylem vessels of the plant. Furthermore, *R. solanacearum* infection has been shown to alter rhizosphere compounds that in turn promote the growth of *Fusarium*
*sp.* [17]. Given such overlap in conditions for the growth of the two pathogens and host infection and their ability to interact in vitro, we deemed it likely that they can interact in the context of disease development, reminiscent of the reported co-infections of the pulmonary system [18,19].

Using tomato as the common susceptible host, we find that disease severity is significantly modulated by co-infection with the two wilt pathogens. Following previous reports, we find that single infections of *R. solanacearum* result in more rapid wilting and increased severity than single infections of *F. oxysporum f.*
*sp.*
*lycopersici* (hereafter referred to as *Fol*). In mixed infections, bacterial wilt severity and bacterial burden in the shoot vasculature are significantly reduced compared to *R. solanacearum* single infections. We show that the severity of wilt disease is dependent on the arrival time of the two microbes at the root infection court. In cases of co-infection where both microbes arrive simultaneously or when *Fol* reaches the root first, *R. solanacearum* wilt severity and *R. solanacearum* recovery from stem tissues are significantly reduced. We further find that bikaverin, an antibacterial metabolite produced by *Fol,* contributes to the reduction in bacterial success in co-infections. Lastly, we identify several uncharacterized secreted secondary metabolites from co-infected tomato rhizosphere that protect tomato seedlings from *R. solanacearum* wilt.

## 2. Results

### 2.1. Bacterial Wilt Severity Is Significantly Reduced in Co-Infections

To study the consequences of co-infection, we developed a co-infection model to simultaneously inoculate ‘Moneymaker’ tomato plants with both *R. solanacearum* and *Fol* (Appendix A). Co-infection resulted in significantly reduced bacterial wilt disease 9 days post infection (dpi) compared to *R. solanacearum* single infections (Figure 1A,B; two-way repeated measures ANOVA *p* < 0.0001; Tukey’s Rs vs. Rs+Foxy *p* < 0.0001). Colonization by *R. solanacearum* was also significantly reduced in co-infected stem tissues (Figure 1C; Mann Whitney U test *p* < 0.0001) demonstrating that the decrease in disease severity is associated with a reduced number of *R. solanacearum* cells in the shoot xylem. Interestingly, co-infected plants showed all-or-nothing wilt phenotypes, i.e., co-infected plants either completely succumbed to *Ralstonia* invasion and wilted completely or showed no wilt 9 dpi (Figure 1A,B) and correspondingly much lower colonization by *Ralstonia* (Figure 1D; ANOVA *p* = 0.0011; Tukey’s multiple comparisons *p* < 0.01). In contrast to the impact of co-infection on bacterial invasion, disease symptomology in co-infected plants over 48 dpi mirrored that of single infections with *Fol* (Figure 1E; ANOVA *p* = 0.0001; Tukey’s Foxy vs. Rs+Foxy *p* = 0.9993) with no differences in fungal colonization between co-infections and *Fol* single infections (Figure 1F; Mann–Whitney U test *p* = 0.3610). Co-culture of *R. solanacearum* and *Fol* in vitro do not show loss of bacterial viability 24 h post inoculation (Appendix A). This indicates that the reduction in *Ralstonia solanacearum* colonization during host co-infection is not due to loss of bacterial viability in the coinfection inoculum. Together, the data show that bacterial wilt severity is reduced in co-infections.

### 2.2. Co-Infection Disease Phenotype Requires Specific Interactions with Live F. oxysporum f. *sp.* lycopersici

To gain insight into how *Fol* contributed to a reduction in bacterial wilt disease during co-infection, we first asked if the live fungus was needed. When *R. solanacearum* was co-inoculated with dead *Fol* conidia (heat killed by exposure to 60 °C for 6 h, verified by plating), bacterial wilt was more severe than in *R. solanacearum* single infections (Appendix A) (two-way repeated measures ANOVA *p* = 0.0005; Tukey’s test Rs vs. Rs + dead Foxy *p* = 0.0227).

We then asked if the interactions were specific to *F. oxysporum f. *sp.* lycoperisici* that can invade tomato xylem. To test this hypothesis, we co-infected tomato plants with *F. oxysporum f. *sp.* tulipae* (Fot), a strain non-pathogenic to tomatoes but pathogenic to tulips. Co-infection of *R. solanacearum* with *F. oxysporum f. *sp.* tulipae* did not reduce bacterial wilt severity (Appendix A) (two-way repeated measures ANOVA *p* = 0.0005; Tukey’s test Rs vs. Rs + Fot *p* = 0.2136). Together, the results suggest that interactions specific to *F. oxysporum f. *sp.* lycopersici* are required for the reduction in bacterial wilt severity.

### 2.3. Early Interactions Play Vital Roles in Co-Infection Disease Outcome

To understand the mechanisms underlying the reduction in bacterial wilt in co-infections, we tested if the arrival time of the pathogens impacted disease severity during co-infection. Figure 2A and Appendix A show the infection timelines for *R. solanacearum* and *Fol* single infections, respectively, assembled from previously published works (references for *R. solanacearum* [20,21,22,23,24,25] and *Fol* [26,27] are indicated in the figures). *R. solanacearum* infection at the root begins with recognition of and irreversible attachment to root hairs, secondary and lateral root emergent sites. This is followed by intercellular invasion into the root cortex past the epidermis, intracellular colonization of cortical cells, finally leading up to systemic invasion and colonization of the plant vascular system resulting in wilting symptoms. Adhesion of *R. solanacearum* cells to the root was not significantly different in simultaneous co-infection compared to *R. solanacearum* single infection (Appendix A).

In order to delineate the relevant time points, *R. solanacearum* was given a headstart of 1 h, 5 h, and 24 h. Even after a 1h headstart, bacterial wilt severity was rescued in co-infections (Figure 2B; two-way repeated-measures ANOVA *p* < 0.0001; 1 h headstart-Rs+Foxy vs. Rs *p* = 0.6131; 1h-headstart-Rs+Foxy vs. Rs+Foxy *p* = 0.0001). Head starts of 5 h and 24 h also resulted in co-infection wilt severity comparable to *R. solanacearum* single infections (data not shown). With a 1h headstart, bacterial CFUs recovered from coinfections were similar to numbers recovered from *R. solanacearum* single infections (Figure 2C; ANOVA *p* < 0.0001; Tukey’s 1h-headstart-Rs+Foxy vs. Rs *p* = 0.7123). On the other hand, a 1h headstart to *Fol* resulted in significantly lower disease compared to simultaneous co-infection (Figure 2D; two-way repeated-measures ANOVA *p* < 0.0001; Tukey’s Rs+Foxy vs. 1 h-headstart-Foxy+Rs *p* = 0.0006) and reduced bacterial burden (Figure 2C; Tukey’s Rs+Foxy vs. 1 h-headstart-Foxy+Rs *p* < 0.000). A 1 h headstart to *Fol* in co-infections also resulted in lower disease compared to *Fol* single infections (Figure 2D; two-way repeated-measures ANOVA *p* < 0.0001; Tukey’s Foxy vs. 1 h-headstart-Foxy+Rs *p* = 0.0095). These results strongly suggest that early bacterial-fungal interactions determine the bacterial fate in planta and the consequent infection outcome.

### 2.4. A Role for Bikaverin in Bacterial Wilt Suppression

Previous work from our lab has shown that the lipopeptide ralsolamycin produced by *R. solanacearum* induced the synthesis of bikaverin in *Fusarium* spp. including *Fol* in in vitro co-cultures [28]. Bikaverin, a polyketide secondary metabolite showed potent antibiotic activity against *R. solanacearum* activity in vitro [28]. Further, our results show that co-infections of *R. solanacearum* with *Fot* do not alter bacterial wilt severity. Interestingly, a protein BLAST analysis with the bikaverin polyketide synthase from *F. fujikuroi* (XP_023430071.1) as query returned no significant similarity matches in the Fot genome. Based on these results, we hypothesized that bikaverin production could play a role in the reduction of bacterial wilt disease during co-infections. To address this hypothesis, *bik1* (encoding the polyketide synthase required for bikaverin synthesis) was deleted in *F. oxysporum f. *sp.* lycopersici* (Appendix AA–C). Co-infection of *R. solanacearum* with ∆*bik1 Fol* unable to produce bikaverin (Appendix AD) resulted in a partial rescue of bacterial wilt (Figure 3A) (two-way ANOVA time x treatment *p* < 0.0001; Tukey’s test: Rs + WT Foxy vs. Rs + ∆*bik1* Foxy *p* = 0.0002; WT Foxy vs. ∆*bik1* Foxy *p* = 0.9994) and bacterial burden (Figure 3B) (Mann–Whitney U test *p* = 0.0472), indicating the contribution of bikaverin to a reduction in bacterial wilt and colonization in co-infection.

### 2.5. Secondary Metabolites in Early Co-Infection Secretome Reduce Bacterial Growth In Vitro

As bikaverin synthesis only partially contributed to the co-infection bacterial wilt suppression and the arrival time of each microbe significantly altered disease during co-infection, we hypothesized that additional early interactions in the rhizosphere played a significant role in the co-infection outcome. As a way of capturing the root co-infection environment, we collected the total rhizosphere secretome from the tri–trophic interaction. The total secretome would contain plant root exudates, as well as secondary metabolites, proteins, lipids and sugars secreted by both microbes and the plant. Studies of plant response in single-pathogen–host systems have identified changes in the composition of root exudates in response to *R. solanacearum* [28]*, Fol* [29] as well as other pathogens [30].

Total growth of *R. solanacearum* in the secretomes collected from each treatment (*R. solanacearum* single infection, *F. oxysporum* single infection, *R. solanacearum-F. oxysporum* co-infection, and uninfected control) was measured as the area under the growth curves (AUC) over 48 h. Comparison of AUCs showed a significant reduction in growth of *R. solanacearum* in the co-infection secretome compared to secretomes from single infections (Figure 4A; ANOVA *p* = 0.0003; Tukey’s Rs vs. Rs+Foxy *p* = 0.0048; Tukey’s Foxy vs. Rs+Foxy *p* = 0.0002). *R. solanacearum* did not show reduced viability upon co-culture in vitro in the buffer used to collect total secretome (Appendix A). We tested bacterial growth in co-infection secretomes that were heat inactivated (100 °C for 5 min; 60 °C for 1 h) where bacterial growth reduction was maintained (Appendix A). These results suggested that the active molecule(s) in the secretome were not proteins but may be primary or secondary metabolites.

To further characterize the potential bioactive molecules, we performed organic solvent extractions from the secretome, eluting lipids into hexane, sugars, carbohydrates, and some secondary metabolites into a methanol/water mixture and the majority of secondary metabolites into ethyl acetate phases based on their polarities. The total co-infection crude extract from the hexane phase did not show activity against *R. solanacearum* (Appendix AA) whereas the methanol (Appendix AB) and ethyl acetate phases (Figure 4B; ANOVA *p* < 0.0001; Tukey’s Rs vs. Rs+Foxy *p* = 0.0005) showed a significant reduction in bacterial growth. Crude ethyl acetate extract *Fol* single infections also showed a reduction in bacterial growth (Figure 4B; Tukey’s Foxy vs. Rs *p* = 0.0025). We focused our efforts on the ethyl acetate phase to identify small molecules involved in the co-infection disease outcome. We only explored the co-infection crude extract since only the co-infection secretome reduced bacterial growth in vitro. Untargeted metabolomics performed on the ethyl acetate crude extracts showed that the secretome of co-infections and single infections are different from each other (Appendix AA). Twenty-one ions were exclusively enriched only in the co-infection secretome compared to *R. solanacearum* or *F. oxysporum* single infections or the uninfected control, while five ions were exclusively diminished (Appendix AB). Several of the ions enriched in co-infection returned no annotation matches using the mzCloud and Chemspider databases (Appendix AC,D) suggestive of novel metabolites that may be upregulated in co-infections.

To specifically identify which secondary metabolites contribute most to the reduction in bacterial growth, we collected 20 fractions of the co-infection crude extract from the ethyl acetate phase and tested for activity against *R. solanacearum* growth in vitro. Five fractions were further chosen for chemical analyses as they showed the highest bacterial growth reduction and were purified in enough quantities for downstream analyses (Figure 4C; ANOVA *p* < 0.0001; Bonferroni correction for multiple comparisons *p* < 0.05). Chromatograms of these five fractions are shown in Figure 4D. LC-MS/MS analyses identified ions with no significant matches in any database (Appendix A). All fractions when applied to tomato roots immediately after infection with *R. solanacearum* provided protection against bacterial wilt (Figure 5A,B; Brown–Forsythe and Welch ANOVA *p* < 0.001; Dunnett’s multiple comparisons *p* < 0.02).

## 3. Discussion

Bacteria and fungi co-occur in large numbers in soil. It is estimated that a gram of soil contains up to 1 billion bacteria and 1 million fungi [31]. Bacteria and fungi are often co-isolated from in-field samples [32] and physically associate with one another in the soil environment [33,34]. Despite the close association of these two Kingdoms in the rhizosphere, surprisingly few efforts have been made to understand the consequences of mixed infections in plant disease. This is in contrast to the increasing number of studies addressing mixed bacterial–fungal infections of humans [2,35]. Our previous studies have identified endofungal behavior of *R. solanaceaum* in fungal species including *Fol* in in vitro co-cultures where the fungus over-produces bikaverin in response to bacterial invasion [14,36]. In soil microcosms, *R. solanacearum* derives overwintering advantages in association with *Fol.* Further, the two pathogens overlap in the range of hosts they infect, share infection routes (via roots to the plant xylem) and cause very similar wilting symptoms in infected plants. Given the remarkable similarities in their infections and their ability to interact in vitro*,* we considered it prudent to examine the consequences of co-infections by these pathogens. We find that upon co-infection of tomato plants, depending on the temporal dynamics of pathogen arrival at the roots, bacterial wilt severity and bacterial load in the shoot xylem are reduced. We show that bacterial wilt suppression is associated with secondary metabolites, both known (bikaverin) and unknown. The uncharacterized natural products show some promise as biopesticides as they protect tomato plants against *R. solanacearum* wilt.

Our results clearly demonstrate that co-infection reduces bacterial wilt severity (Figure 1). Further, early interactions (up to ~24 h) between *R. solanacearum, Fol* and the susceptible tomato host are strong determinants of the infection outcomes (Figure 2). This emphasizes the need for studying early infection events and early intervention efforts for effective management of wilt disease. We also find that the live fungus is required for reduction in bacterial wilt during co-infection (Appendix A) and that *Fusarium* host specificity may be important (Appendix A). Whether the difference lies in the ability of the fungus to simply invade the roots, cause wilting symptoms, or induce certain plant defense responses could possibly be delineated by examining coinfections with other fungal wilt pathogens such as *Verticillium* spp. [37] or different strains of *Fusarium oxysporum* such as the tomato endophyte *F. oxysporum* Fo47 [38]. Our study also identifies a role for the antibacterial secondary metabolite bikaverin in bacterial disease suppression (Figure 3) and supports our previous results showing inhibition of *R. solanacearum* growth by this metabolite [28]. However, bikaverin presence could not fully account for disease suppression leading us to analyze the infection secretome.

Investigation into the early cross-kingdom secretome at the root identified putative secondary metabolites secreted during co-infection that restricted *R. solanacearum* growth in vitro (Figure 4A,B). Fractionation of crude extracts from the co-infection secretome identified several fractions that reduced *R. solanacearum* growth in vitro with novel ions identified with LC-MS/MS analyses (Figure 4C,D; Appendix A). In planta evaluation of five of these fractions showed protection to a susceptible tomato host from *R. solanacearum* disease (Figure 5). Bikaverin was not found in the rhizosphere secretome likely due to the early time point at which the metabolome was assessed. It is also possible that bikaverin is more important in later stages of the infection process.

The chemical structures of the secretome metabolites, the producing organism synthesizing these natural products, and their antibacterial mechanisms remain unknown. Future work characterizing the putative novel compounds identified here can evaluate their efficacy in controlling *R. solanacearum* wilt. Current strategies for managing bacterial wilt emphasize prevention as once *R. solanacearum* invades a field the bacterium survives for long periods of time, sometimes up to four years [39]. The primary focus is on the use of resistant cultivars, clean planting and harvest equipment, and quick removal of infected plants from the field as currently available chemical and biological control is often ineffective against *R. solanacearum* [40]. Therefore, new and robust management strategies are a constant need to manage disease. Our results demonstrate that exploring interactions between microbes that share infection niches and routes may be powerful ways of identifying novel bioactive small molecules for disease management. This is an expansion of the already realized power of eliciting the production of bioactive secondary metabolites from in vitro bacterial–fungal co-cultures [41].

A comprehensive model of interaction between *R. solanacearum* and *Fol* in soil and in the presence of a susceptible host emerges from our results (Figure 6). This work characterizes co-infection outcomes by two devastating plant pathogens and identifies secondary metabolites at the interface of the cross-kingdom communication, unveiling potentially new biopesticides against *R. solanacearum* wilt.

## 4. Materials and Methods

### 4.1. Microbial Culture Maintenance and Growth Conditions

All fungal and bacterial strains used in the study and their sources are listed in Appendix A. All strains were stored at −80 °C in 33% glycerol. *R. solanacearum* was routinely grown in CPG medium, amended with 0.005% Tetrazolium chloride, and incubated at 30 °C for 2 days. Colonies from freshly grown plates were cultivated overnight in liquid CPG broth at 30 °C with constant shaking at 250 rpm. *F. oxysporum f. *sp.* lycopersici* and *F. oyxposrum f.* sp*. tulipae* were routinely grown on PDA for 5 days. Plugs were then transferred to 50 mL of PDB in 125 mL flasks and incubated at 30 °C for 5 days with constant shaking at 250 rpm to raise conidia. For secondary metabolite extractions from fungal cultures, fungal plugs from PDA plates were grown in 50 mL of liquid ICI media in 125 mL flasks and incubated at 30 °C for 5 days with constant shaking at 250 rpm.

### 4.2. Media Recipes

CPG (1L): 1 g Casamino acids, 1 g Bacto™ Yeast Extract, 10 g Dextrose, 10 g Bacto™ Peptone, 16g agar.

Boucher’s Minimal Medium (BMM 1 L): 3.4 g KH_2_PO_4_, 0.5 g (NH_4_)_2_SO_4_, 100 µL of 1.25 mg/mL stock solution FeSO_4_ 7H_2_O, 517 µLof 1 M stock solution MgSO_4_, 2 g Dextrose, pH 6.5.

PDB/PDA (Potato Dextrose Broth/Agar): Difco™ PDB and PDA used as indicated in the label.

ICI (1L): 10 g Dextrose, 0.48 g NH_4_NO_3_, 5 g KH_2_PO_4_, 1 g MgSO_4_.7H2O, 1 mL 1000× trace elements.

1000× trace elements (100 mL): 2.2 g ZnSO_4_.7H_2_O, 1.1 g H_3_BO_3_, 0.5 g MnCl_2_.4H_2_O, 0.5 g FeSO_4_.7H_2_O, 0.16 g CoCl_2_.5H_2_O, 0.16 g CuSO_4_.5H_2_O, 0.11 g (NH_4_)_6_Mo_7_O_24_.4H_2_O, 5 g Na_4_EDTA.

### 4.3. Strain Construction and Transformation

The predicted sequence of the backbone polyketide synthase *bik1* ortholog in *Fol* was identified based on nucleotide and protein BLAST search with *bik1* characterized in *F. fujikuroi* (FFUJ_06742) as the template. Sequence alignment identified homologous regions in genes FOXG_04757 and FOXG_04758 (Appendix AA). Both putative genes aligned with *bik1* of *F. fujikuroi* indicating they are both part of *bik1* of *Fol.* Thus, the two genes FOXG_04757 and FOXG_04758 have been annotated as *bik1* in the rest of the text and Figureures. The WT *Fol* was used as the parent strain to delete *bik1* with the homologous recombination-based integration of the hygromycin resistance gene *hph* at the *bik1* locus. *hph* was amplified from the plasmid pAN7-1. The schematic representation of the *bik1* gene replacement with *hph* is shown in Appendix AB. The three amplified fragments (5′ flank upstream of *bik1, hph* from plasmid pAN7-1, 3′ flank downstream of *bik1*) were fused together to form the deletion cassette with the double-joint PCR described elsewhere.

Transformation of *Fol* protoplasts was performed based on the protocol below to generate the deletion strains (denoted as 2.5 and 2.C in Figure 6C). The reagents required for transformation include; PDB, KCl/CaCl_2_ buffer (1.2 M KCl, 50 mM CaCl_2_), STC solution (0.8 M sorbitol, 50 mM CaCl_2_, 50 mM Tris-HCl, pH 7.5), TEC solution (10 mM Tris-HCl, pH 7.5, 1 mM EDTA, 40 mM CaCl_2_), PEG solution (60% (w/v) polyethylene glycol MW 4000 in 0.6 M MOPS). The regeneration medium used to plate protoplast–DNA–PEG mixture includes MgSO_4_ 7H_2_O (0.5 g/L), KH_2_PO_4_ (1 g/L), KCl (0.5 g/L), NaNO_3_ (2 g/L), glucose (20 g/L), sucrose (200 g/L), oxoid agar (14 g/L for regeneration plates and 5 g/L for top agar).

Spores (5 × 10^8^) were inoculated in 100 mL PDB at 28 °C, 250 rpm for 12–16 h. Germlings were harvested with centrifugation and washed with KCl/CaCl_2_ buffer (concentration mentioned in the previous paragraph) twice before resuspension in the enzyme cocktail (200 mg Lyzing Enzyme (Sigma), 150 mg Driselase (Fluca), 15 mg Lyticase (Sigma), 10 mg Yatalase (Takara) in 50 mL KCl/CaCl_2_ buffer). The mixture was incubated for 2 h at 28 °C at 100 rpm and routinely examined for protoplasts. When ready, protoplasts were collected by filtering through a double-layered Miracloth. This was followed by centrifugation and washing with two volumes of STC. The protoplasts were then pelleted with centrifugation and resuspended in STC. Next, 5–10 μg of the transforming DNA was mixed with TEC solution to a final volume of 60 μL. About one hundred microliters of protoplasts was added to TEC solution and incubated on ice for 20 min. One hundred and sixty microliters of PEG was mixed in and incubated at room temperature for 15 min. One milliliter of STC was then added. The mixture was centrifuged to pellet protoplasts and resuspended in STC. The protoplasts were spread onto a plate containing the regeneration medium and incubated at 28 °C overnight. Hygromycin B was then overlaid at 100 µg/mL followed by incubation for 3–5 days.

Transformants were picked as hygromycin-resistant single colonies growing out of hygromycin amended plates containing the protoplasts and the deletion cassette. All transformants were screened by PCR and a single insertion at the *bik1* locus was confirmed with a Southern blot using [α32P] dCTP (PerkinElmer, USA) to label the DNA probes (schematic in Appendix AB and results in Appendix AC). The primers used for mutant construction and screening are listed in Appendix A. Lack of bikaverin production by the deletion mutant was confirmed with metabolite extraction and HPLC analysis (Appendix AD).

### 4.4. Tomato Coinfection Procedures

Conidia of *Fol* used for infection were collected from 5 to 6-day old liquid cultures as described earlier. Overnight cultures of *R. solanacearum* were grown to OD > 1 to obtain bacterial cells for infection. Bacterial cells and fungal conidia were washed twice in sterile double distilled water and mixed together in flasks at a ratio of 1:1000 (2.5 × 10^5^ cells of the bacterium and 2.5 × 10^8^ conidia of *Fol* in 30 mL) to obtain the co-inoculum to perform co-infections. Inoculum of either the bacteria alone or the fungal conidia alone was used to perform single infections. 18 to 20-day old tomato plants of the cultivar ‘Moneymaker’ (from ParkSeed) were uprooted from existing pots, rinsed in running water three times, and then dipped well-mixed in single or co-inocula for 3 min. These plants with dripping roots were then transplanted into healthy unautoclaved soil and monitored for plant disease (Appendix A). The plants were grown under consistent and controlled conditions at 28 °C and a 12 h photoperiod. Plants were watered with water or Hoagland’s solution alternatively. Disease severity values were assigned to each individual plant based on the percentage of leaves that showed wilting phenotype:1 (1–25% wilting), 2 (26–50%), 3 (51–75%), and 4 (76–100%).

### 4.5. Assessment of Bacterial Adhesion

Four-day-old tomato seedlings germinated on 1% water agar were laid out on a new water agar plate (Appendix AA). A 2 cm region of the root was marked and 10 µL of co-inoculum (10^4^
*R. solanacearum* cells/mL and 10^7^
*Fol* conidia/mL) was applied along the length. The infected plants were incubated for 2 h at 28 °C. The 2 cm portion was excised, washed thoroughly with agitation, and pressed in between sheets of filter paper. The root pieces were homogenized and plated on CPG amended with 130 µg/mL cycloheximide to select for bacterial colony growth.

### 4.6. Assessment of Microbial Load in Planta

At 9 days post infection (dpi), ~0.1 g of stem 1–2 cm above the cotyledons were excised, homogenized, and dilution-plated on bacterial selective (CPG + 130 µg/mL cycloheximide) or fungal selective (PDA + 100 µg/mL ampicillin) media. Bacterial and fungal colonies were normalized to the corresponding individual tissue weights.

### 4.7. Rhizosphere Secretome Growth Analyses

Six to seven-day-old tomato seedlings germinated on 1% water agar were used. Thirty seedlings were pooled per technical replicate to account for plant-to-plant variation. The seedlings were placed in 5 mL of chemotaxis buffer [20] in 50 mL conical tubes along with single or co-inocula. *R. solanacearum* cells were inoculated to a final concentration of 5 × 10^4^ cells/mL and *Fol* conidia to a final concentration of 5 × 10^7^ conidia/mL. The setup was incubated for 24 h after which the buffer along with the secreted metabolites were collected and filtered through a 0.2 µm filter. The secretome collected was plated to confirm the lack of microbial cells. *R. solanacearum* growth over time was assessed with growth curves over 42–60 h as appropriate in 96-well plates. The crude extracts and fractionated crude extracts were resuspended in an 80:20 acetonitrile–water mixture. This mixture was added to a final concentration of 5% in Boucher’s Minimal Medium and appropriate controls were used. Optical density at 600 nm was read at 1 h intervals. The area under each growth curve was calculated as a measure of overall bacterial growth.

### 4.8. Metabolite Extraction and Fractionation

Secondary metabolite extraction from *Fol* pure cultures was performed as described in Spraker et al. 2018 [28]. Metabolite extraction from the secretomes started with the addition of equal volumes of methanol and hexane to the collected aqueous secretome in a separating funnel. After mixing, the hexane layer separated (top layer) and contained eluted fats and oils. The collected methanol layer (bottom layer) was poured back into the separating funnel. This was topped off with ethyl acetate (ethyl acetate:methanol in a 4:1 ratio). A few drops of water were added until the layers separated. The aqueous layer (bottom layer) and the ethyl acetate layer (top layer) were collected separately. All phases (hexane, methanol, or ethyl acetate) were evaporated under reduced pressure and each crude extract was reconstituted in 80:20 acetonitrile:water at 200 mg/mL by dry weight of the crude extract. Fractionation of the co-infection crude extract from the ethyl acetate phase was accomplished with a preparatory HPLC (details in the following section) and fractions were collected every 2 min. The fractions were evaporated and reconstituted in an 80:20 acetonitrile:water mixture at 100 mg/mL.

### 4.9. Analytical Chemistry

HPLC-DAD was performed on a Gilson GX-271 Liquid Handler with a system 322 H2 Pump connected to a 171 Gilson Diode Array Detector. The XBridge C18 3.5 µm 4.6 × 150 mm column was used for the analytical run with a flow rate of 0.8 mL/min. HPLC grade water with 0.5% formic acid (solvent A) and HPLC grade acetonitrile with 0.5% formic acid (solvent B) were used with the gradient: 0 min, 20% Solvent B; 2 min, 20% Solvent B; 15 min, 95% Solvent B; 20 min, 95% Solvent B; 20 min, 20% Solvent B; 25 min, Solvent B. An XBridge BEH C18 OBD Prep Column (130 Å, 5 µm, 19 mm × 250 mm) with an XBridge BEH C18 Prep Guard Cartridge (130 Å, 5 µm, 19 mm × 10 mm) was used for preparative HPLC with a flow rate of 16 mL/min. HPLC grade water with 0.5% formic acid (solvent A) and HPLC grade acetonitrile with 0.5% formic acid (solvent B) were used with the following gradient 0 min, 20% Solvent B; 2 min, 20% Solvent B; 15 min, 95% Solvent B; 20 min, 95% Solvent B; 20 min, 20% Solvent B; 25 min, Solvent B. Data acquisition and analysis were performed with TRILUTION LC V3.0.

UHPLC-HRMS was performed on a Thermo Scientific-Vanquish UHPLC system connected to a Thermo Scientific Q Exactive Orbitrap mass spectrometer in ES+ and ES- mode between 200 m/z and 1400 m/z to identify metabolites. A Zorbax Eclipse XDB-C18 column (2.1 × 150 mm, 1.8 μm 123 particle size) was used with a flow rate of 0.2 mL/min. LC-MS grade water with 0.5% formic acid (solvent A) and LCMS grade acetonitrile with 0.5% formic acid (solvent B) were used with the gradient: 0 min, 20% Solvent B; 2 min, 20% Solvent B; 15 min, 95% Solvent 126 B; 20 min, 95% Solvent B; 20 min, 20% Solvent B; 25 min, Solvent B. Data acquisition and analysis for the UHPLC-MS were performed with Thermo Scientific Xcalibur software Version 3.1.66.10. The acquired data were also analyzed with Compound Discoverer 3.2 to generate plots Appendix A.

### 4.10. Evaluation of Extract Efficacy against Bacterial Disease

To evaluate whether the fractions from co-infection crude extract protected tomato plants from *R. solanacearum* disease, an infection model was developed on Hoagland’s medium. 6-day old seedlings germinated on 1% water agar were transferred to Hoagland’s agar and incubated at 28 °C with a 12 h photoperiod for 7 days. The plants were then infected by applying *R. solanacearum* suspensions of 5 × 10^6^ cells/mL as 1 µL droplets at each point covering the root. After the inoculum dried, 1% of 100 mg/mL fractionated extracts were applied along the roots, also as 1 µL droplets covering the root. One percent of an 80:20 acetonitrile:water mixture was used as a solvent control. The plants were incubated at 28 °C with a 12 h photoperiod for 6 days before symptoms were recorded.

### 4.11. Statistical Analyses

All statistical analyses were performed with GraphPad Prism version 8.3.0. Normal distribution was tested with Shapiro–Wilk’s test. The area under the curve was calculated with the default parameters on GraphPad Prism. Repeated measures ANOVA and post hoc analyses, t-tests, and non-parametric t-tests were performed as appropriate for the experiments and are detailed in the figure legends.

## Figures and Tables

**Figure 1 microorganisms-09-02123-f001:**
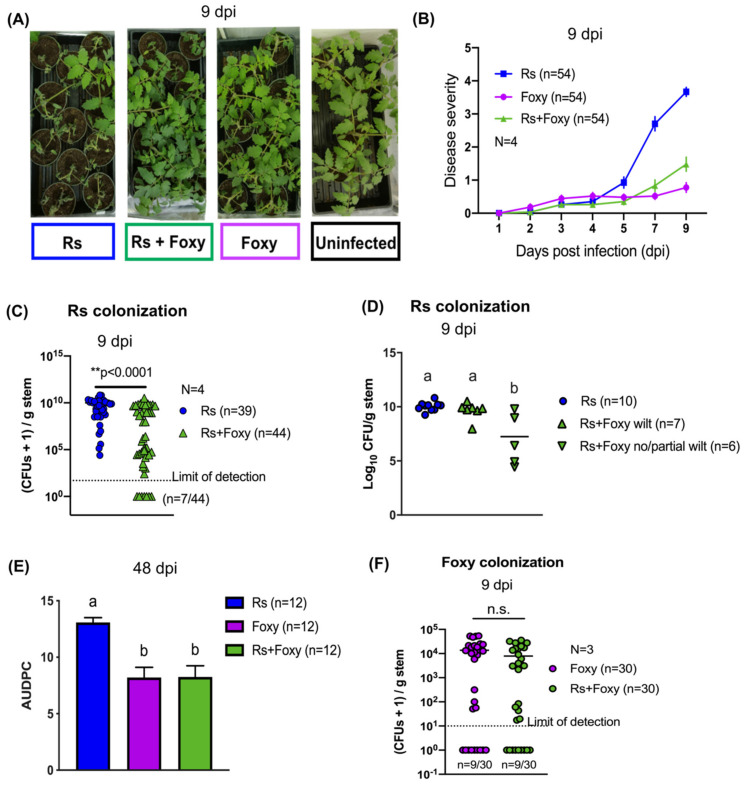
Co-infection reduces bacterial wilt severity. (**A**) Single-infected and co-infected showing differential wilting symptoms 9 days post infection (dpi). *R. solanacearum =* Rs; *F. oxysporum* (Foxy). (**B**) Wilt symptoms were scored every day for 9 dpi and quantified. Two-way repeated-measures ANOVA was performed with time x treatment *p* = 0.0001. Tukey’s multiple comparisons showed significantly different disease between Rs vs. Foxy (*p* < 0.0001) and Rs vs. Rs+Foxy (*p* < 0.0001). No significant difference in symptoms was observed between Foxy and Rs+Foxy (*p* = 0.7867). (**C**) Bacterial colony forming units (CFUs) quantified 9 dpi from shoots 1 cm above the cotyledons showed reduced bacterial burden in co-infection compared to single *R. solanacearum* infections. All CFU data in this work is presented as CFUs+1 to take into account samples below the detection limit, denoted under each dataset. Mann–Whitney’s U test was performed with *p* < 0.0001. (**D**) Co-infected plants that fully wilted had bacterial burden equal to plants infected with only *R. solanacearum* (ANOVA *p* = 0.0011; Tukey’s *p* = 0.8259). Co-infected plants that showed no/partial wilt symptoms had bacterial burden lower than *R. solanacearum* single infections (Tukey’s *p* = 0.0011). (**E**) Disease was quantified over 48 days based on disease incidence. ANOVA was performed on AUDPC with *p* = 0.0001. No significant difference in overall disease was observed between co-infection and *F. oxysporum* single infections (Tukey’s *p* = 0.9993). (**F**) *F. oxysporum* CFUs quantified from shoot tissues 9 dpi show no significant differences in fungal burden. Mann–Whitney’s U test *p* = 0.3610.

**Figure 2 microorganisms-09-02123-f002:**
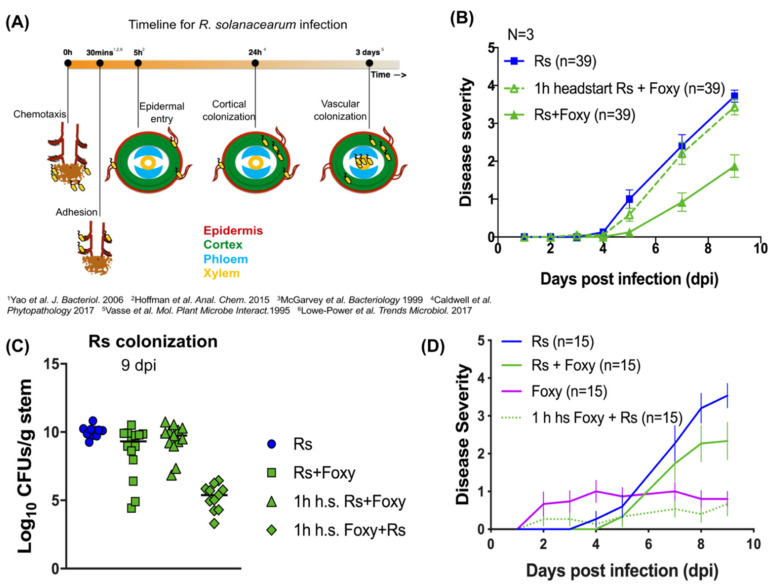
Early interactions determine co-infection outcome. (**A**) timeline for *R. solanacearum* infection from recognition of the host until colonization of the xylem, gathered from available literature. (**B**) The figure shows disease severity over 9 days for 1 h head-start to *R. solanacearum* in co-infection (1 h headstart Rs + Foxy) with relevant controls. Two-way repeated-measures ANOVA was performed with time x treatment *p* < 0.0001. A 1 h head-start to *R. solanacearum* in co-infection resulted in disease equal to *R. solanacearum* single infection (1 h headstart Rs+Foxy vs. Rs—Tukey’s *p* = 0.6131). Simultaneous co-infection resulted in reduced disease (Rs+Foxy vs. 1 h headstart Rs+Foxy—Tukey’s *p* = 0.0001). (**C**) CFUs of *R.* solanacearum from shoot tissue 9 dpi; ANOVA *p* < 0.0001. While co-infection resulted in reduced colonization (Rs vs. Rs+Foxy Tukey’s *p* = 0.0452), a 1 h head-start to *R. solanacearum* resulted in similar colonization as found in bacterial single infections (Rs vs. 1h h.s. Rs+Foxy Tukey’s *p* = 0.7123). (**D**) shows disease severity over 9 days for 1 h head-start to *F. oxysporum* in co-infection (1 h h.s. Foxy + Rs) with relevant controls. Two-way repeated-measures ANOVA was performed with time x treatment *p* < 0.0001. A 1 h headstart to *F. oxysporum* in co-infection resulted in lower disease than simultaneous co-infection (Rs+Foxy vs. 1 h h.s. Foxy+Rs—Tukey’s *p* = 0.0006).

**Figure 3 microorganisms-09-02123-f003:**
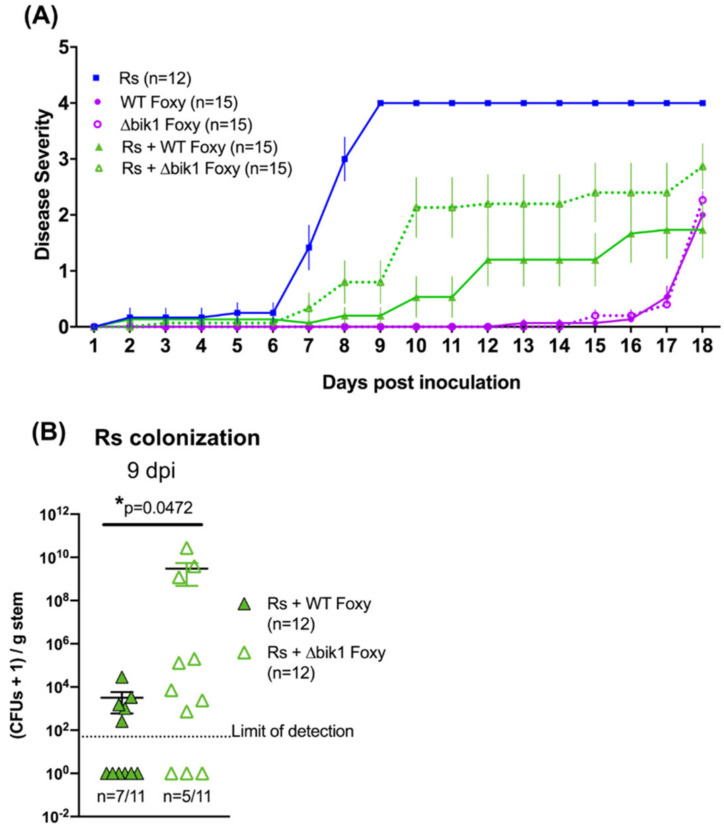
Bikaverin contributes to the reduction in bacterial wilt in co-infections. (**A**) Disease severity scores were assigned daily for 18 days. ‘∆bik1 Foxy’ refers to the *F. oxysporum* strain whose backbone polyketide synthase gene (*bik1)* required for bikaverin production was deleted. Two-way repeated-measures ANOVA was performed with time x treatment *p* = 0.0001. Tukey’s multiple comparisons showed significantly different disease between Rs + WT Foxy and Rs + ∆*bik1* Foxy (*p* = 0.0002), Rs vs. Rs + WT Foxy (*p* < 0.0001), and Rs vs. Rs + ∆*bik1* Foxy (*p* < 0.0001). No significant difference in symptoms was observed between WT Foxy and ∆*bik1* Foxy (*p* = 0.9994). (**B**) Bacterial CFUs recovered 9 dpi from shoots 1 cm above the cotyledons from plants co-infected with *R. solanacearum* and WT or ∆*bik1 F. oxysporum* strains. CFUs are presented as CFUs+1 to take into account samples below detection limit, denoted nder each dataset. The data was non-normal. Mann–Whitney’s U test was performed with *p* = 0.0472.

**Figure 4 microorganisms-09-02123-f004:**
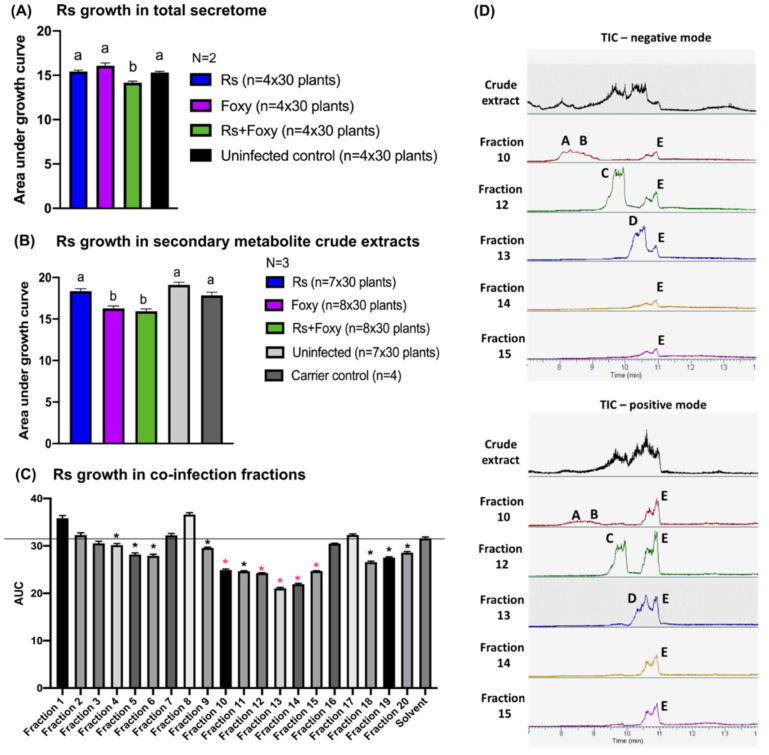
Crude extracts from the co-infection secretome reduce in vitro *R. solanacearum* growth. (**A**) shows area under growth curve (AUC) for 42 h post inoculation (hpi). *R. solanacearum* was inoculated in total secretome (including plant and microbial secretions) from different treatments as indicated. Bacterial growth in co-infection secretome was significantly reduced compared to secretomes from single infections (ANOVA *p* = 0.0003; Rs vs. Rs+Foxy—Tukey’s *p* = 0.0049; Foxy vs. Rs+Foxy—Tukey’s *p* = 0.0001). (**B**) shows *R. solanacearum* growth as AUC up to 48 hpi in crude extracts eluted into ethyl acetate. ANOVA *p* < 0.0001. Rs vs. Rs+Foxy—Tukey’s *p* = 0.0005; Foxy vs. Rs+Foxy—Tukey’s *p* = 0.9260. (**C**) Co-infection crude extract was fractionated and *R. solanacearum* growth was quantified in each fraction. ANOVA *p* < 0.0001. Bonferroni correction was performed to compared AUCs between each fraction with the solvent control **p* < 0.05. The fractions marked in pink * (*p* < 0.0001) were chosen for further analyses in (**D**) and Figure 4. (**D**) Total ion chromatograms (TIC) of the fractions in positive and negative modes run on HR-ESI-MS. The m/z of these peaks (**A**–**E**) are reported in Appendix A.

**Figure 5 microorganisms-09-02123-f005:**
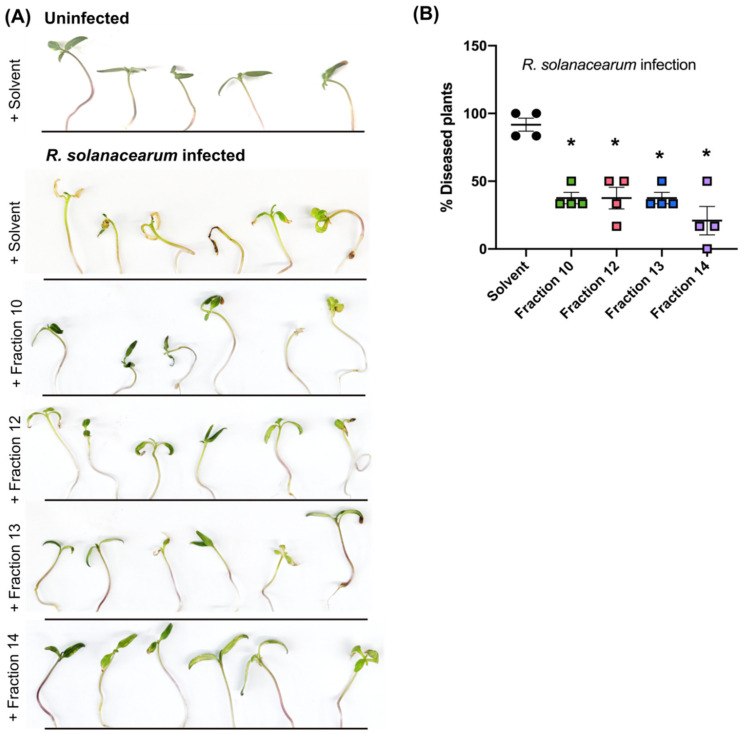
Secondary metabolites from co-infection secretome protect tomato seedlings from *R. solanaceaum* wilt. (**A**) 11-day old seedlings were infected with *R. solanacearum* after which either 80% acetonitrile (solvent control) or fractions of the ethyl acetate crude extract were applied to the roots at a final concentration of 1% by volume in water. The top panel shows an uninfected control for reference. The plants were photographed 6 dpi. (**B**) Four individuals scored each plant as to whether they were infected or not. The % diseased plants reported by each person is shown in the graph. ANOVA *p* < 0.0001. Bonferroni correction was performed to compare disease between solvent and fractions. In all cases, * *p* < 0.0003.

**Figure 6 microorganisms-09-02123-f006:**
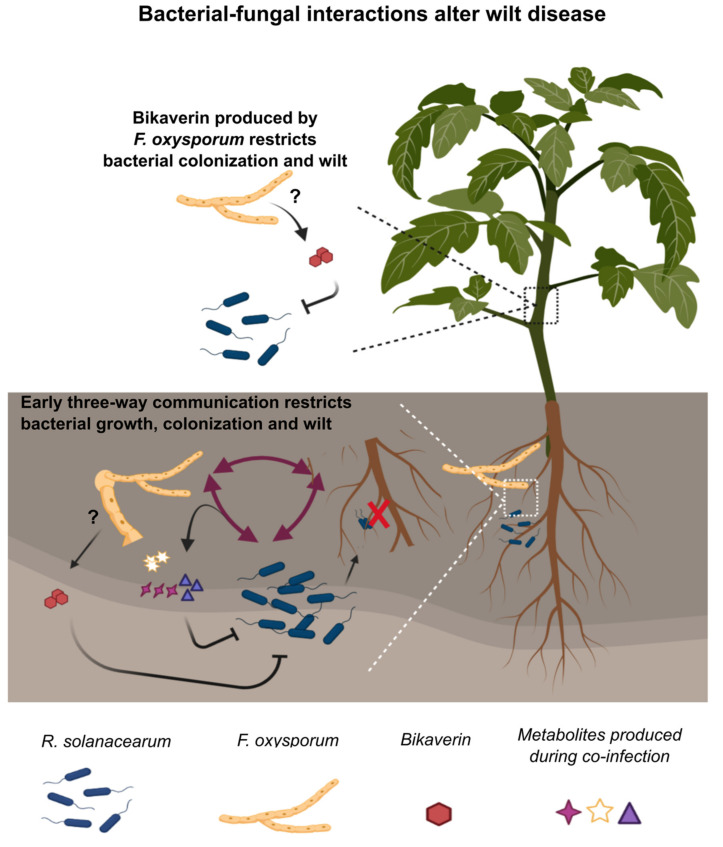
Model for tri–trophic interactions mediating wilt disease. Early interactions between the microbes and the host at the root result in the production of secondary metabolites that inhibit *R. solanacearum* growth and colonization. In the presence of the bacterium, *F. oxysporum* produces bikaverin, whose antibacterial activity also contributes to reduced colonization by *R. solanacearum.* Whether bikaverin is produced in the rhizosphere or in later stages inside the plant remains to be answered (represented with a question mark). This figure was created with Biorender.com.

## Data Availability

All data is available in the main text or the Appendix A. All strains will be provided from the Keller Lab upon request.

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
