# Peer review of "Secreted Secondary Metabolites Reduce Bacterial Wilt Severity of Tomato in Bacterial–Fungal Co-Infections"

_microorganisms, 2021, doi:10.3390/microorganisms9102123_

Round 1

Reviewer 1 Report

The manuscript from Venkatesh et al describes in detail how the inter kingdom coinfection with Ralstonia solanacearum (Rs) and Fusarium oxysporum (Foxy) in Tomato plants reduces bacterial wilt severity, and that Bikaverin synthesis by Fusarium contributes to bacterial wilt reduction. Further, analysis has been done on the co-infection root secretome to identify previously uncharacterized secreted metabolites which probably contribute to the protection of tomato seedlings against bacterial wilt disease. The Manuscript is very well written and supported by well-done experiments in detail to derive certain scientific conclusion(s).

However, I have some concern on the set up of Tomato Coinfection Procedures from which certain fundamental conclusions have been drawn in this Manuscript. As the timeline for bacterial and fungal infection events on tomato roots are quite different in natural environment (Figure 2A and Figure S4), the idea of studying the effects of co infection on tomato roots under this nonnatural experimental set up might not reflect  the real scenario. Moreover, for co-inoculum “ratio of 1:1000 (2.5X105 cells of the bacterium and 2.5x108 conidia of Fol)” was used to perform coinfection. Why was this 1:1000 ratio chosen? Why not 1:1? In natural root rhizosphere of wilted tomato plant, what is the load of individual microbes and spores? Related to Tomato wilting, is inter kingdom co-infection tend to be a general occurrence in nature? In this manuscript plants were inoculated for disease symptom by dipping the roots directly for 3 minutes in single or co-inocula. Why were this technique chosen with a relatively high microbial load in planta? Why was it not inoculated indirectly in sterile soil or inert surface in pots where the plants are growing which might better simulate natural environment?

Su et al., 2020 have previously reported that Rs invasion increased the concentrations of three phenolic acids in the rhizosphere soil of bacterial wilt-diseased tomato plants than those in the rhizosphere soil of healthy tomato plants which consequently stimulated F. solani growth. Was there any indication of enhancement of phenolic acids also in Rs-Foxy coinfection of tomato? Is it also likely that Rs preferentially stimulate Foxy growth in Rs-Foxy coinfection?

Lastly if the hypothesis that bacterial wilt severity is reduced in co-infection with Foxy, what is its evolutionary significance in terms natural selection? This part might be discussed in the Manuscript.

Reviewer 2 Report

Komentarze do Autorów

UWAGI OGÓLNE

Dość miejsce jest manuskrypt zatytuÅ‚owany: „Wydzielane metabolity wtórne redukujÄ… nasilenie bakteryjnego wiÄ™dniÄ™cia w koinfekcjach bakteryjno-grzybowych”. Badania te wyjaÅ›niajÄ…, co skutkuje powstaniem choroby mieszanych.

Unfortunately, in my opinion, not all chapters of the manuscript were written correctly. The introduction is interestingly written. It has well-chosen content. The authors used a correct research methods and received interesting results and is the value of this manuscript. Statistical data analysis is at a sufficient level. In my opinion, the abstract should be slightly edited. The chapter on results and discussion requires the most extensive revision. I my opinion, in this manuscript a short chapter of conclusions is extremely needed, because it is difficult to understand the final achievements of the research from the description of the results.

SPECIFIC COMMENTS

Abstract

The abstract should be written concisely and present the most important research results. The wording "taken together" is not indicated in this part of the manuscript. (lines 24-25, page 1).  

“Results” chapter

In my opinion, the results are described incorrectly. The content of the chapter should be rewritten. Elements of the hypotheses and solutions searches should not appear, but synthetically described research results. This way of describing the results does not give an opportunity to understand what results of the research were finally obtained. (lines 111 -125, page 4)

The description also includes undesirable elements of the research methods used. (line 119-121, page 4)

This is not a professional description of the results. This is not a professional description of the results. Is it the description of the results of this study or the course of the infection described by other authors? (line 129-135, page 4)

If the described results are not presented, we rather avoid describing them. It is certainly not professional to add the words "date not shown" in brackets. (line 144, page 4)

“Previous work from our lab has shown that the lipopeptide ralsolamycin produced by R. solanacearum induced the synthesis of bikaverin in Fusarium spp. including Fol in  in vitro co-cultures [28]. Bikaverin, a polyketide secondary metabolite showed potent antibiotic activity against R. solanacearum activity in vitro [28].”

This is an excerpt from the discussion, which was also repeated in the "discussion" section. The results should be synthetically described, excluding elements of the discussion. The results should be fully edited in this regard. (lines 169-172, page 5)

“As bikaverin synthesis only partially contributed to the co-infection bacterial wilt suppression and the arrival time of each microbe significantly altered disease during co-infection, we hypothesized that additional early interactions in the rhizosphere played a significant role in the co-infection outcome. As a way of capturing the root co-infection  environment, we collected the total rhizosphere secretome from the tri-trophic interaction.” (lines 196-200, page 7)

This should be part of the discussion.

lines 214-216 , page 7 - This should be part of the discussion.

wiersze 236-236, s. 8 Zdania te są bardziej do metodologii badań

RozdziaÅ‚ „Dyskusja”

Z kolei w reakcji na zmiany wyników. W tej sekcji separuje siÄ™ od ustawienia do sekcji. Podam odniesienia do fragmentów, które wymagajÄ… przeredagowania.

linie 286 -288, s. 10

wiersz 291, strona 10

wiersz 297, s. 10

wiersze 301-307, strona 10 - TÄ™ część etykiety należy prawować, ponieważ jest to opis wyników.

wiersze 324 -328, strony 11-12 Ta część opisu i krótkiej drogi prowadzÄ…cej

Wnioski

Moim można powiedzieć, że w tym manuskrypcie jest krótki okres jest niezmiernie gwarantowany warunek, że niemożliwe jest osiÄ…gniÄ™cie ostatecznej dorobku badaÅ„.
